# Molecular Aspects of Piperine in Signaling Pathways Associated with Inflammation in Head and Neck Cancer

**DOI:** 10.3390/ijms25115762

**Published:** 2024-05-25

**Authors:** Juliana Prado Gusson-Zanetoni, Luana Pereira Cardoso, Stefanie Oliveira de Sousa, Laura Luciana de Melo Moreira Silva, Júlia de Oliveira Martinho, Tiago Henrique, Eloiza Helena Tajara, Sonia Maria Oliani, Flávia Cristina Rodrigues-Lisoni

**Affiliations:** 1Department of Biology, Institute of Biosciences, Humanities and Exact Science (IBILCE), São Paulo State University (UNESP), São José do Rio Preto 15054-000, Brazil; juliana.gusson@unesp.br (J.P.G.-Z.); luana.cardoso@unesp.br (L.P.C.); stefanie.sousa@unesp.br (S.O.d.S.); llmm.silva@unesp.br (L.L.d.M.M.S.); julia.martinho@unesp.br (J.d.O.M.); sonia.oliani@unesp.br (S.M.O.); 2Department of Molecular Biology, School of Medicine of São José do Rio Preto (FAMERP), São José do Rio Preto 15090-000, Brazil; henrique@famerp.br (T.H.); tajara@famerp.br (E.H.T.)

**Keywords:** inflammation, herbal medicine, PTGS2, MAPK, MMPs, cytokines

## Abstract

Piperine, an active plant alkaloid from black pepper *(Piper nigrum*), has several pharmacological effects, namely antioxidant, anti-inflammatory and immunomodulatory effects, which involve inhibiting molecular events associated with various stages of cancer development. The aim of this study was to investigate the molecular mechanisms of action of piperine in relation to its potential anticancer effect on head and neck cancer cells. Parameters related to neoplastic potential and cytokine, protein and gene expression were investigated in head and neck cancer cell lines (HEp-2 and SCC-25) treated with piperine. The results of the tests indicated that piperine modified morphology and inhibited viability and the formation of cell colonies. Piperine promoted genotoxicity by triggering apoptosis and cell cycle arrest in the G2/M and S phases. A decrease in cell migration was also observed, and there was decreased expression of MMP2/9 genes. Piperine also reduced the expression of inflammatory molecules (PTGS2 and PTGER4), regulated the secretion of cytokines (IFN-*γ* and IL-8) and modulated the expression of ERK and p38. These results suggest that piperine exerts anticancer effects on tumor cells by regulating signaling pathways associated with head and neck cancer.

## 1. Introduction

Cancer is promoted by genomic instability that affects cell growth, metabolism and inflammation, and this has been associated with higher rates of recurrence and mortality in head and neck cancer (HNC) [1]. These tumors are malignant and develop in the facial, oral and neck regions, affecting the upper aerodigestive tract, salivary glands and thyroid. This type of cancer ranks sixth among the most common cancers worldwide and is associated with high mortality due to intervening in vital life functions such as phonation, swallowing, breathing, taste and smell [2].

Each year, 450,000 global deaths are associated with HNC, and it is considered a clinically heterogeneous disease that involves different risk factors and differences in molecular pathogenesis. In addition to the two major risk factors, tobacco and alcohol consumption, oncogenic viruses, the human papillomavirus (HPV), the microbiome and diet have also been established in recent decades as contributing sources for the development of this disease [3]. Treatment for patients with HNC depends on the site of origin of the tumor and generally includes surgical resection, radiotherapy, chemotherapy, molecular therapy, immunotherapy and the use of natural products as an adjuvant modality [4].

Several natural products affect various oncogenic signaling pathways simultaneously, modulating the activity or expression of their molecular targets. These include cell death by apoptosis, proliferation, migration/invasion and angiogenesis. Thus, natural products are capable of generating intracellular signals that trigger events leading to the death of cancer cells. One of the most important sources of biologically active compounds is the plant kingdom, so there is a large list of phytochemicals (chemical compounds produced by plants) with therapeutic activity, including terpenes, alkaloids, essential oils, flavonoids and primary and secondary metabolites [5]. Among these phytochemicals, we can mention piperine (1-piperoylpiperidine), which is an alkaloid derived from plants of the Piperaceae family and can be isolated mainly from the fruits or roots of black pepper (*Piper nigrum*) and long pepper (*Piper longum*) [6].

Alkaloids are a class of chemical compounds derived from natural sources, characterized by a cyclic structure and the presence of a central nitrogen atom. Most heterocyclic rings consist mainly of nitrogen atoms [7]. The structure of piperine comprises functional groups identified by the formation of reactive metabolites, namely a methylenedioxyphenyl (MDP) ring and conjugated alkene and piperidine portions [8]. Furthermore, the piperine molecule contains an oxymethylene ring, an aliphatic olefin chain and an amide group [9].

Piperine is used to increase the bioavailability of several drugs due to its capacity to modulate the P-glycoprotein and cytochrome P450 systems. This mechanism occurs precisely due to the barriers of the chemical structure of piperine, which gives it the ability to interact with several pathways involved in the pathogenesis of diseases [10]. This compound is biosynthesized from L-lysine and the precursor cinnamoyl-CoA as a secondary metabolite. After a series of reactions, the bioactive compound piperine is obtained from piperonyl-CoA (Figure 1) [11,12].

The attention given to the study of this molecule is mainly due to its biological properties, such as anti-inflammatory, antioxidant, immunomodulatory and anticancer properties, which allow this compound to chemically interact with various molecular targets [7]. Specifically, in relation to piperine’s anticancer activity, recent observations have shown that its mechanism of action is multiple and involves the activation of cell signaling pathways, such as cell proliferation, programmed cell death and decreased migration and invasion of cancer cells [13].

In this context, studies indicate that piperine can modulate various molecular targets, such as receptors and enzymes (prostaglandin E2 receptors, cyclooxygenase 2 and matrix metalloproteinases (MMPs)), kinases (including the mitogen-activated protein kinase (MAPK) pathway, ERK1/2 and p38), inflammatory cytokines (IL-1β, IL-2, IL-8 and IFN-γ), inflammatory mediators such as JNK, AP-1 (activator protein 1), iNOS (nitric oxide synthase) and gene expression modulators (miRNAs) [14,15,16].

The effects and possible molecular mechanisms of action of piperine on cancers including head and neck carcinoma have been recently studied, but not completely elucidated, mainly in relation to inflammation in laryngeal and tongue cancer [16,17]. Therefore, the aim of this study was to investigate the effects of piperine on the signaling pathways that modulate the molecular mechanisms of cancer-associated inflammation, which could be used as a therapeutic alternative in this type of cancer.

## 2. Results

### 2.1. Piperine in High Concentrations Modifies Morphology, Reduces Viability and Causes Cytotoxicity in Head and Neck Cancer Cell Lines

Cell morphology changes were observed in HEp-2 and SCC-25 cells after treatment with piperine. Photomicrographs of the control groups showed morphology typical of HEp-2 and SCC-25 cells (Figure 2A,C), while the shape of the cells treated with piperine showed changes such as shrinkage and decreased cell-to-cell contact (Figure 2B), and the formation of clusters, detachment from the surface and decreased cell density, respectively (Figure 2D).

With regard to cell viability analysis, it was evident in HEp-2 cells (Figure 2E) that treatment with piperine at concentrations of 150, 200, 250 and 300 μM decreased viability at 24, 48 and 72 h. For SCC-25 cells, a reduction in cell viability was observed at 48 h, specifically at concentrations of 100, 150, 200 and 250 μM of piperine, while at the other times of 4, 24 and 72 h, piperine did not reduce the viability of these cells when compared to the DMSO control group (Figure 2F).

As for cytotoxicity, the IC_50_ of piperine varied according to the exposure time of the compound in the cells studied, with the HEp-2 cells (102.8 to 176.0 µM) and the SCC-25 cells (121.0 to 249.9 µM), as shown in Appendix A. Together, the proliferation and viability/cytotoxicity tests revealed that the effective concentration and treatment time of the compound in the cells was 150 µM in 24 h of action since at higher concentrations, piperine is considered highly toxic to cells.

### 2.2. Piperine Has an Antiproliferative and Cytostatic Effect on Head and Neck Cancer Cell Lines

The results showed that in comparison with untreated cells, piperine inhibited the growth of HEp-2 and SCC-25 cells, with the concentrations of 200 and 300 μM showing the greatest effect after 24 h of treatment in the two cell lines evaluated (Figure 3A). In relation to the colony formation assay of HEp-2 and SCC-25 cells, it was observed that piperine decreased the ability to multiply the growth of cell colonies and also the potential related to the number of cells per colony, when compared to the DMSO control group (Figure 3B).

### 2.3. Piperine Induces Apoptosis and Cell Cycle Arrest in Head and Neck Tumorigenic Cells through Genotoxicity

Our results indicated that piperine induced significant apoptosis in both cell lines studied. Treatment of HEp-2 and SCC-25 cells with piperine resulted in approximately 26.5% and 22% of early and late apoptosis, respectively (Figure 4A). With regard to verifying cell cycle arrest, the results showed that piperine caused a significant accumulation of cells in the G2/M phase in the HEp-2 lineage, and in parallel, there was a decrease in these cells in the G0/G1 and S phases. For the SCC-25 cells, this retention mechanism occurred in the S phase of the cycle, thus promoting cell DNA synthesis (Figure 4B).

Figure 5 shows the migration of fragmented DNA caused by treatment with piperine. The average damage index of the control groups of HEp-2 and SCC-25 cells was 62.3 and 78.6, respectively, and after treatment with piperine, this damage jumped to 137.3 and 159, thus showing that the treatment generated genotoxicity for the cells analyzed.

### 2.4. Piperine Decreases Cell Invasion by Reducing the Rxpression of Metastasis-Related Genes in Head and Neck Cancer Cells

Our findings indicate that piperine decreases the invasion capacity of HEp-2 and SCC-25 cells (Figure 6A), which suggests piperine’s anti-invasive activity in head and neck carcinoma cells. The HEp-2 cells in the control group had an average migration rate of 70.3, and in the group of cells treated with piperine, this figure was 12.3. For the SCC-25 lineage, piperine also reduced this migratory mechanism, as the group of treated cells had numerical indications of 10.3, while in the control group, this average was 24.6.

In view of piperine’s action against cell invasion, we sought molecular proof of the expression of genes related to metastasis. The genes analyzed using the PCRq technique in the HEp-2 and SCC-25 cell lines were *MMP2* and *MMP9* (Figure 6B). Treatment with piperine in HEp-2 cells significantly decreased the expression of the *MMP2* and *MMP9* genes. However, for the SCC-25 strain, the significant decrease only occurred for the *MMP2* gene, and the *MMP9* gene was not differentially expressed in this strain. We also checked the expression of the MMP2 protein in HEp-2 and SCC-25 cells (Figure 6C), and the results showed no statistically significant difference in the expression of this protein in the two strains evaluated. We only observed a downward trend in MMP2 protein expression for the SCC-25 cell. In this respect, piperine did not modulate this enzyme at the gene translation level.

### 2.5. Piperine Regulates the Expression of Genes, Cytokines and Proteins Associated with Inflammation

The inflammatory response of the cells was evaluated according to the expression levels of *PTGS2* and *PTGER4* after treatment with piperine (Figure 7A). Piperine reduced the expression levels of *PTGS2* and *PTGER4* in HEp-2 cells. For the SCC-25 strain, *PTGS2* expression was also reduced, while the *PTGER4* gene was not modulated and was not differentially expressed.

The concentrations of the cytokines IL-8, IL-1β and IFN-γ released in the culture supernatants were measured by ELISA. The release of IL-8 and IFN-γ was considerably decreased in response to piperine treatment compared to cells from the DMSO control group in the HEp-2 strain (Figure 7B). However, treatment with piperine did not significantly alter IL-1β expression compared to control cells in this cell type. For SCC-25 cells, the secretion of the cytokines IL-8, IL-1β and IFN-γ was significantly reduced after treatment with piperine when compared to untreated cells.

In addition, the protein activity of PTGS was observed; our results indicated a significant decrease in the expression of this protein after treatment with piperine in the SCC-25 strain, and in HEp-2 cells, no significant results were found for the expression of this enzyme (Figure 7C).

### 2.6. Piperine Modulates the Expression of the ERK/p38 MAPK Pathway in Head and Neck Cancer Cells

To better determine piperine’s anti-inflammatory mechanism, MAPKs (ERK and p38) were examined in HEp-2 and SCC-25 cells. The results showed that piperine inhibited the expression of ERK and p38, indicating a significant reduction in these proteins compared to the control group. It is also important to note that the inhibition of ERK and p38 expression occurred in all the cell compartments analyzed (nucleus, cytoplasm and total), in both strains evaluated (Figure 8).

## 3. Discussion

The identification of new anticancer therapeutic agents is a fundamental issue for the study and development of drugs aimed at treating this disease. Among these agents, we can highlight piperine, an alkaloid derived from *Piper nigrum*, which has anti-inflammatory, antioxidant, and immunomodulatory effects [18]. One of the main biological characteristics involving inflammatory/carcinogenic processes is the capacity for increased cell proliferation. In the present study, the antiproliferative action of piperine was verified, including the inhibition of the multiplication of colonies in the cells studied. Other in vitro studies have also shown this antiproliferative effect of piperine in colon, lung, breast and hepatocellular adenocarcinoma cell lines, as a result of inducing cell cycle arrest in the G1 phase and regulating the expression of p21/WAF1 and p27/KIP1 [19,20].

Cell viability and cytotoxicity are other important indicators in the in vitro toxicological evaluation of a given compound [21]. Our analyses indicated that piperine decreases cell viability upon application of the treatment, and it was even pointed out that the main biological effects of piperine in vitro occur at specific doses (75–200 µM) and an incubation time between 24 and 48 h, thus confirming the short-term exposure data for piperine [22]. This cytotoxicity effect is described as piperine promoting the inhibition of NADH-oxidoreductase, an enzyme that stimulates cell activity and proliferation, as well as disruption of mitochondrial membrane permeability [23].

Piperine also induced apoptosis and cell cycle arrest in the G2/M and S phases in the cells studied. These mechanisms have also been described in several studies, such as in DU145 prostate cancer lines and SNU-16 and GES-1 gastric cancer lines, probably due to a decrease in the expression of anti-apoptotic proteins (Bcl-2 and Bcl-xl), which initiate caspase signaling that is responsible for the destruction of cell structure and consequent apoptotic death. Therefore, the induction of apoptosis and the arrest of the cell cycle are the main mechanisms of studies related to the discovery of compounds with possible activities against cancer [24,25].

Furthermore, many chemotherapeutic drugs are genotoxic agents and induce apoptosis, due to the generation of DNA damage, as does piperine in high concentrations [26]. The antioxidant capacity of tumor cells can be deregulated by an increase in reactive oxygen species (ROS). Alkaloids, such as piperine, with pH-dependent ionizable groups, can bind and interact with DNA, and this interaction generates breaks in the DNA chain, compromising the integrity of genetic information [27]. Recently, Rezaei and collaborators (2024) discovered that piperine increases lipid peroxidation and decreases the activity of superoxide dismutase, and this may occur due to the binding of this compound to the minor groove of DNA [28], and in another study, it was found that this connection occurs due to piperine reaching the G-quadruplex structure of DNA [29]. G-quadruplex structures are non-standard forms of DNA that are made in telomeres and regulatory regions of oncogenes, mainly c-myc, when cells divide. These structures are extremely important in preserving shortened telomeres and facilitating genomic instability, thus promoting tumor growth. The potential of piperine may be linked to the stability of the G-quadruplex structure [30]. So, in our study, we analyzed the damage index generated by the treatment and thus confirmed that piperine caused DNA fragmentation in the cells studied.

In addition to the antiproliferative and genotoxic activity of piperine, the anti-metastatic effect is another efficiency observed in some studies with this herbal medicine. This action of piperine is seen in triple-negative breast cancer cells (MDA-MB-468, T-47D and MCF-7) and colorectal cancer cells (SW480 and HCT-116), by decreasing the mRNA expression of metalloproteinase 2 and 9 [31,32]. In our experiments, we found that this invasive mechanism in HEp-2 and SCC-25 cells was significantly inhibited after treatment with piperine, probably suggested by the discovery of the modulation of MMP2 and MMP9 mRNA expression in these cells, as well as the decrease in cell growth observed in the proliferation assay, a mechanism also seen in another study [33].

In order to better understand the mechanisms of cancer progression, some mediators (PTGS2 and PTGER4), which have been extensively studied by our research group previously, have been verified [34,35]. In addition, tumorigenic cells have a characteristic of uncontrolled growth, and it is known that as arachidonic acid derivatives participate in inflammation and are also closely linked to tumor development, with PTGS2 being highly expressed in hyperplastic tissues, the study of the genes of this inflammatory pathway is necessary in head and neck cancer [36]. Our results on gene expression indicate that piperine significantly reduced *PTGS2* levels in the cells studied. A significant decrease in *PTGER4* expression occurred in the HEp-2 lineage, and this result was not observed in the SCC-25 cells. In terms of protein expression, our findings showed that the treatment reduced PTGS2 in both cells studied, indicating that piperine modulates gene and protein expression in our cancer model. Thus, our data corroborate the results of Kim and collaborators (2012), who also found a marked decrease in the expression levels of PTGS2 genes and proteins in mouse macrophage cells after treatment with piperine [37].

Among the main mediators of inflammation are cytokines and proteins belonging to the mitogen-activated protein kinase (MAPK) family. Interfering with chronic inflammation means interfering with the function of these mediators and the signal transduction dependent on these molecules [38]. In this respect, the secretion of the cytokines studied, IL-8, IL-1β and IFN-*γ*, and the expression of MAPKs (ERK and p38) were reduced following treatment with piperine, thus indicating a further mediating effect of this compound on HEp-2 and SCC-25 cells. Thus, our data corroborates other results from our research group, in which a decrease in cytokine levels (IL-8, IL-1β) and ERK and p38 was also observed in HeLa, SiHa and CaSki cervical cancer cells after treatment with piperine [35]. Other studies also mention this action of piperine in decreasing ERK and p38 in breast cancer cells [39]; in addition, Western blot results confirmed that piperine decreased the expression of JNK and p38 in human ovarian cancer cells [40].

Thus, piperine exerts its effects by modulating inflammation-mediating molecules such as genes, cytokines and proteins (MAPK, MMPs, PTGS2) via cyclooxygenase 2. In addition, this study provides a new understanding of the role of piperine in molecular events and signaling pathways that are directly related to the development of head and neck cancer.

Considering the use of piperine as an adjuvant product in the treatment of inflammation and tumorigenesis, it would be preferable that this compound was also analyzed in animal models, in order to verify its action at a systemic level, as there are few studies that show the action of piperine in a model in vivo [41]. Therefore, it is still of great importance that more research is carried out to better verify this compound in animal models with pathologies related to cancer [42]. But it is possible to observe that piperine is an effective antitumor compound in vitro and in vivo and has the potential to be developed as a new anticancer drug [43], even in clinical studies; its action has also been found to be anti-inflammatory [44].

Thus, our study has some limitations, such as the lack of experiments with non-tumor cells, as we had great difficulty finding a lineage of the same histological type/tissue as the cancer cells. However, in another study carried out by our research group, it was found that piperine had a less intense effect on normal HaCaT cells. Therefore, our study did not use a normal cell line as a control, due to the previous results observed in the other study. Furthermore, we recognize that it would be important to confirm our findings with functional studies, using pharmacological inhibitors for example, but given the above considerations, we suggest that further research be carried out to elucidate the effects of piperine as a natural and integrative treatment for patients with head and neck cancer.

## 4. Materials and Methods

### 4.1. Cell Lines and Treatment with Piperine

The HEp-2 cell line (laryngeal squamous cell carcinoma) was seeded in MEM-Earle medium and the SCC-25 cell line (tongue squamous cell carcinoma) in DMEM-HAM-F12 (Cultilab, Campinas, SP, Brazil), both supplemented with 10% fetal bovine serum (Cultilab, Campinas, SP, Brasil), 1% antibiotic/antimycotic (Invitrogen, Carlsbad, CA, USA), 1% L-glutamine 200 µM (Sigma-Aldrich, St. Louis, MO, USA), 1% non-essential amino acids 10 mM and 1% sodium pyruvate 100 mM (Sigma-Aldrich, St. Louis, MO, USA), cultivated under standard conditions (37 °C, 5% CO_2_), and sourced from the American cell line bank (ATCC, Manassas, VA, USA). For the initial experiments, piperine (Sigma-Aldrich, St. Louis, MO, USA, P49007) was used for the cells at concentrations of 25 μM, 50 μM, 100 μM, 150 μM, 200 μM, 250 μM and 300 μM, diluted in 0.1% dimethylsulfoxide (DMSO-Sigma, D8418—negative control), at times of 4, 24, 48 and 72 h. After being considered functional for the cells, without presenting a high degree of cytotoxicity, only one concentration (150 µM) and one incubation time (24 h) of piperine treatments were chosen. All tests were carried out in triplicates and in three individual experiments.

### 4.2. Viability/Cytotoxicity Assay and Cell Proliferation

HEp-2 and SCC-25 cells (5 × 10^3^) were evaluated by MTS reagent (Promega, Madison, WI, USA) to determine the viability of the tumor cells in a 96-well plate. Different concentrations of piperine were prepared (25 μM, 50 μM, 100 μM, 150 μM, 200 μM, 250 μM, 300 μM). Each experimental condition received 20 μL of MTS solution (3-(4,5-dimetiltiazol-2-il)-5-(3-carboximetoxifenil)-2-(4-sulfofenil)-2Htetrazólio) (Promega). The optical density was measured at 490 nm in a microplate reader (BioRad, Hercules, CA, USA) at 4, 24, 48 and 72 h. The IC_50_ (50% inhibitory concentration) was defined as the concentration of the sample that reduced the absorbance by 50% compared to the control, using the function in GraphPad Prism 8.0.1 software.

For the growth curve, in 24-well culture plates, density 5 × 10^4^, in 500 μL of complete medium, the concentrations of each treatment used were 100, 200 and 300 μM at 4, 24, 48 and 72 h. Subsequently, the cells were trypsinized, stained with Trypan Blue and counted in the automated cell counter (Countess Automated Cell Counter II, Life Technologies). For cell viability and proliferation, two-way analysis of variance (ANOVA) and Dunnett’s test were applied.

### 4.3. Transwell Invasion Assay

Approximately 5 × 10^4^ cells were added to the upper chamber of the inserts (transwell, BD Biosciences San Jose, CA, USA), along with 200 µL of serum-free medium, and 750 µL of complete medium containing 10% serum was added to the lower compartment of the well. The cells were incubated at 37 °C, 5% CO_2_ for 24 h and then fixed in paraformaldehyde and stained with crystal violet. The insertions were counted and photographed under a microscope (100× magnification)(Olympus, Tóquio, Japão). The numbers of cells that crossed the membrane were counted for statistical analysis using the *t*-test.

### 4.4. Clonogenic Assay

The cells (8 × 10^2^) were seeded in 6-well plates and incubated under the experimental conditions (DMSO control and piperine at a concentration of 150 μM). After 24 h, the medium was changed, and the treatment was added; every 2 days, this condition was renewed over a period of 14 days. The cells (colonies) were then fixed with methanol, stained with crystal violet and counted by visual inspection, and the results were analyzed using the *t*-test.

### 4.5. Determination of Apoptosis and Cellular DNA Content

Cells (1 × 10^6^) after treatment with piperine were analyzed by flow cytometry (Guava Easy Cyte, MILLIPORE) and incubated with fluorochrome-conjugated ANXA5 monoclonal antibody (PE, BD Pharmigen, San Diego, CA, USA) and with 7-ADD. Cell cycle arrest was assessed using the Guava® Cell Cycle Reagent kit (MILLIPORE, USA), using the protocol proposed by the manufacturer. Tumor cells (1 × 10^6^) were washed with PBS, fixed in 70% ethanol, resuspended in the Cell Cycle kit solution and evaluated by flow cytometer (Guava Easy Cyte, MILLIPORE).

### 4.6. Genotoxicity Test (Alkaline Comet)

Cell sediments (5 × 10^4^) were mixed with low-melting-point agarose and placed on slides containing a mixture of PBS (phosphate buffer solution) and normal-melting-point agarose. These slides were subjected to the lysis step and electrophoresed. The slides were then neutralized and fixed in 100% ethyl alcohol. The slides were stained with a solution of Gel Red 10,000×, 1M NaCl and distilled water and analyzed under a fluorescence microscope. The cell nuclei (100 cells per group) were classified into a damage class (0 to 4) and subjected to a formula to determine the total damage index. The statistics were based on the Kruskal–Wallis non-parametric analysis of variance, and the means were compared using the Mann–Whitney test.

### 4.7. Enzyme-Linked Immunosorbent Assay (ELISA)

The expression patterns for each cytokine/chemokine tested (IL-1β, IL-8 and IFN-*γ*) were analyzed according to the manufacturer BD Biosciences. The supernatant of the cells was collected from each experimental group, and then the analyses were read on a spectrophotometer at a wavelength of 450 nm. The data were plotted and analyzed using the *t*-test.

### 4.8. Immunocytochemistry Analysis

The cells were cultured at a concentration of 1×10^5^ on culture slides (Nunc, Naperville, IL, USA) and fixed (4% paraformaldehyde), permeabilized in Triton X, washed with PBS-T and subjected to blocking (PBS+Normal goat serum +BSA). Immunolabeling was performed with primary mouse monoclonal antibodies (Ab) anti-p38/MAPK (BD Bioscience, USA) and anti-ERKpan (BD Biosciences, USA) diluted 1:200, and secondary goat anti-mouse IgG antibody conjugated to Alexa Fluor 546 (Dako, Glostrup, Denmark), for about 1 h. The slides were then mounted (DAPI) for analysis under an Axioskop 2 fluorescence microscope (Zeiss, GR). Ten digital images of each replicate were captured using AxioVision software 4.8, version Axiskope 2 mot plus (Zeiss, GR), where six cells from each image were evaluated by densitometry obtained using Image J version V 1.8.0 (National Institute of Health—NIH, Bethesda, MD, USA). Statistical analysis was performed using the *t*-test.

### 4.9. RNA Isolation, Target Genes and Real-Time PCR Analysis

Initially, total RNA was extracted using Trizol TRI Reagent^®^ (Sigma-Aldrich, St. Louis, MO, USA), followed by reverse transcription in cDNA using the High Capacity cDNA Reverse Transcription Kit (Applied Biosystems, Forster City, CA, USA), as described by the manufacturer. The reaction for the *PTGS2*, *PTGER4*, *MMP2* and *MMP9* genes was carried out in a 7500 Fast Real-Time PCR System thermal cycler (Applied Biosystems), prepared in triplicate and processed in a final volume of 20uL containing 50 ng of cDNA, SYBR® Green PCR Master Mix and 100nM of each primer, according to the Applied Biosystems protocol. The 2^−∆∆Ct^ method [45] was used for relative quantification of gene expression, with the levels of the GAPDH (Glyceraldehyde-3-phosphate dehydrogenase) gene used as internal controls. The primers used were as follows: *PTGS2* (f: 5′ATTCCCTTCCTTCGAAATGC3′; r: 5′ AGAAGGCTTCCCAGCTTTTG3′); *PTGER4* (f: 5′ CGAGATCCAGATGGTCATCTTAC 3′; r: 5′ CCAAACTTGGCTGATATAACTGG 3′); *MMP2* (f: 5′ AAGTCTGGAGCGATGTGACC 3′; r: 5′ CCGTCAAAGGGGTATCCATC 3′); *MPP9* (f: 5′ TTGTGCTCTTCCCTGGAGAC 3′; r: 5′ ATTTCGACTCTCCACGCATC 3′) e *GAPDH* (f: 5′ CTGTTGCTGTAGCCAAATTCGT 3′, r: 5′ ACCCACTCCTCCACCTTTGA 3′).

### 4.10. Expression of Protein Levels (Western Blotting)

Protein concentrations were quantified using the BCA protein assay kit (Thermo Scientific, Wilmington, DE, USA). The expression of PTGS2 (1:500 Abcam, Cambridge, UK) and MMP2 (1:500 ABclonal, Woburn, MA, USA) were examined, and equal amounts of proteins (30 µg) were separated by polyacrylamide gel electrophoresis (Bio-Rad, Hercules, CA, USA) and transferred to nitrocellulose membranes. The membranes were blocked with 5% powdered milk diluted in TBS-T and incubated with the specific primary and secondary antibodies (anti-rabbit IgG—1:1000 Abcam, Cambridge, UK). The endogenous control beta-actin was simultaneously detected in the reaction by the mouse monoclonal anti-igG antibody (Abcam, Cambridge, MA, USA) at a concentration of 1:2000. The protein expression levels obtained were calculated and presented as the mean ± SEM of the mean optical density and subjected to the *t*-test.

## 5. Conclusions

Piperine has anticancer activity due to its anti-inflammatory and antiproliferative properties and could be a prospective and integrative therapeutic option for patients with head and neck cancer.

## Figures and Tables

**Figure 1 ijms-25-05762-f001:**
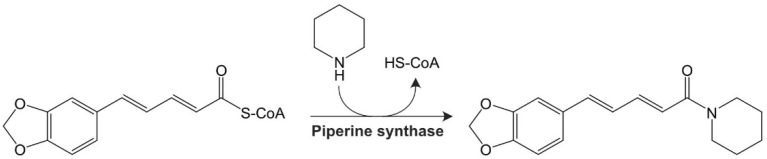
Piperine biosynthetic pathway: piperine synthase catalyzes the formation of the piperine molecule from piperidine and piperoyl CoA.

**Figure 2 ijms-25-05762-f002:**
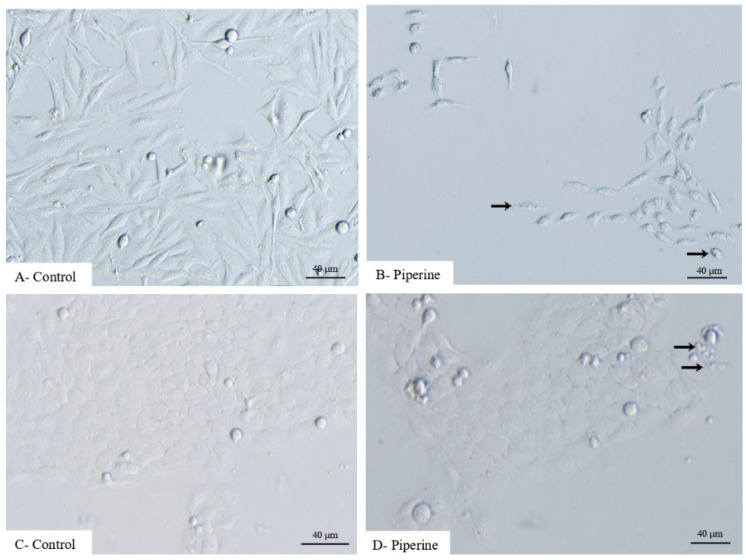
Photomicrographs of the morphology of control (**A**) and piperine-treated (**B**) HEp-2 cells. Control SCC-25 cells (**C**), and cells treated with piperine (**D**), at a concentration of 150 μM, over 24 h, with black arrows pointing to cell shrinkage and the formation of blebs (cellular vesicles). Graphs of cell viability for (**E**) HEp-2 and (**F**) SCC-25, using the 7 concentrations tested (25 μM, 50 μM, 100 μM, 150 μM, 200 μM, 250 μM, 300 μM), at 4, 24, 48 and 72 h. * with *p* < 0.05. *n* = 3. Bars: 40 μM.

**Figure 3 ijms-25-05762-f003:**
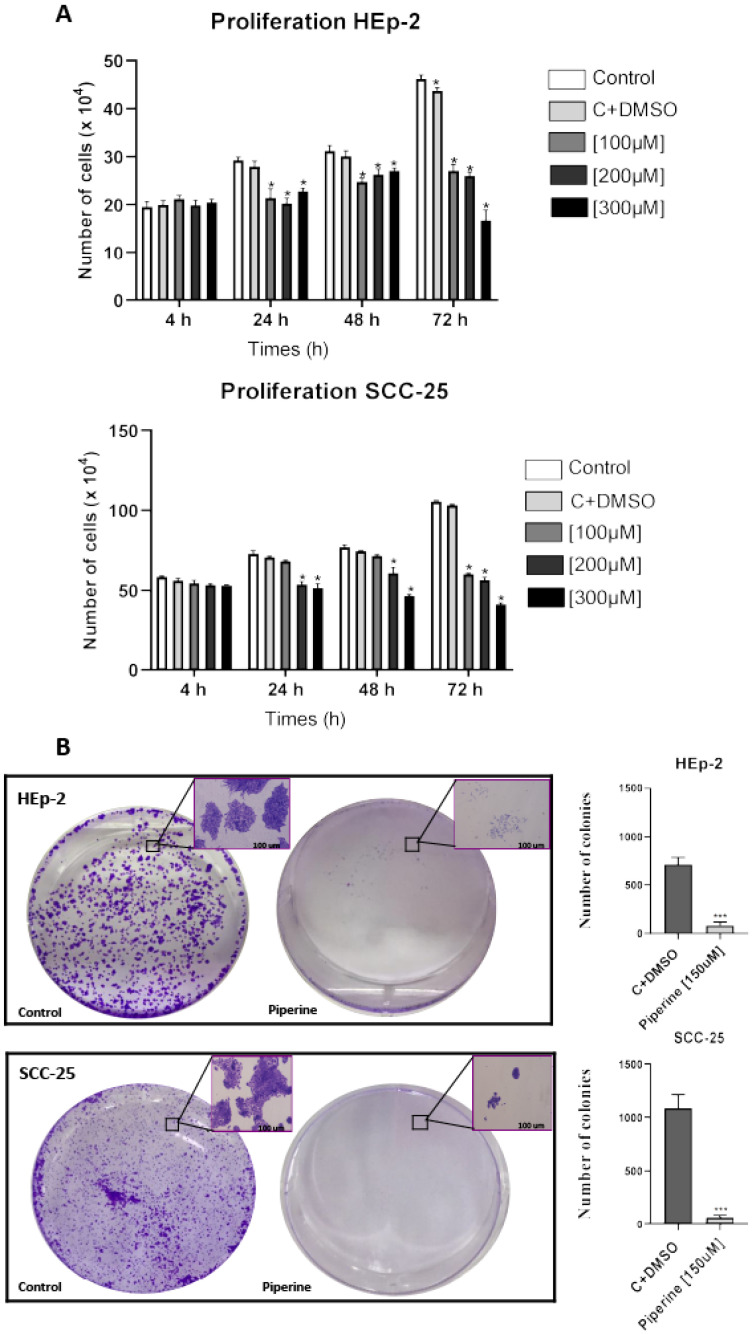
Growth curve of HEp-2 and SCC-25 cell lines (**A**) treated with piperine at three concentrations (100, 200 and 300 μM) for 4, 24, 48 and 72 h. * *p* < 0.05. Photomicrograph of the colony formation assay in HEp-2 and SCC-25 strains (**B**), after treatment with piperine (150 μM), after 24 h, and graphs statistically representing colony formation, with comparisons of the control and piperine groups. *** vs. control, *p* < 0.0001. *n* = 3.

**Figure 4 ijms-25-05762-f004:**
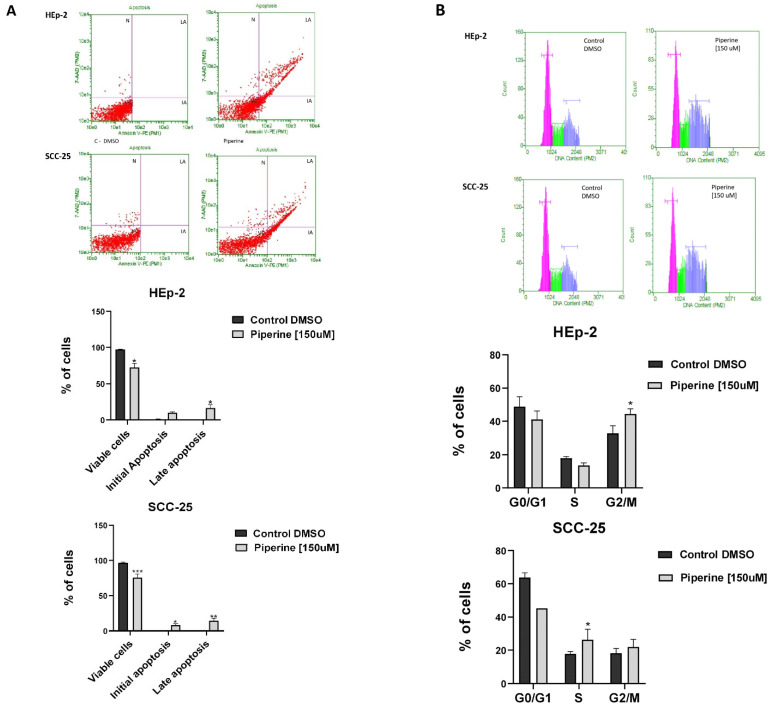
Detection and statistical analysis of apoptosis (**A**). Treatment with piperine at a concentration of 150 μM, and a time of 24 h, significantly increased the rate of apoptosis in HEp-2 laryngeal cancer cells and SCC-25 tongue cancer cells, marked as CV (viable cells), AI (initial apoptosis), AT (late apoptosis) and N (necrosis). Comparison between groups, * *p* < 0.05. Cell cycle analysis (**B**). Treatment with piperine (150 μM over 24 h) promoted cell cycle arrest in the Hep-2 and SCC-25 cancer cell lines. Comparison between groups, * *p* < 0.05. *n* = 3. ** *p* < 0.01; *** *p* < 0.001.

**Figure 5 ijms-25-05762-f005:**
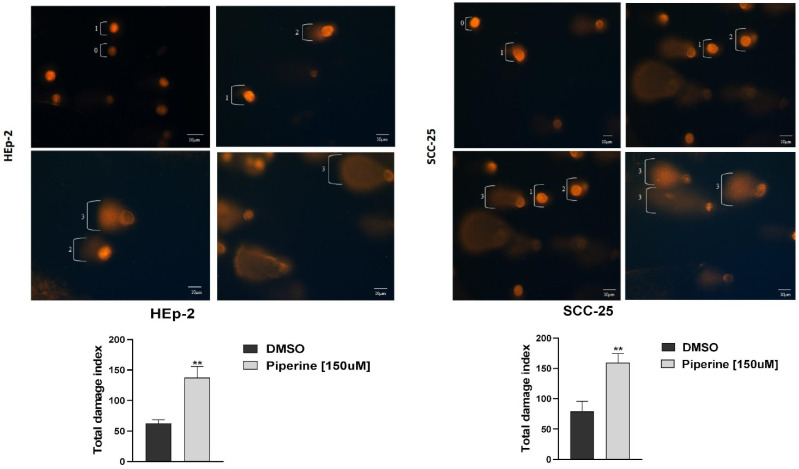
Evaluation of the genotoxicity assay, with fluorescence photomicrographs of the treated group and nuclear damage index of HEp-2 and SCC-25 cells treated with 150 μM of piperine in 24 h, with damage from 0 to 3, where 0 indicates no apparent damage, 1 indicates damage one times the size of the nucleus, 2 indicates damage twice the size of the nucleus, and 3 indicates damage two or more times the size of the nucleus. Graphs depicting comet assay statistics, with ** statistically significant difference between treatment and controls, with *p* ≤ 0.05. *n* = 3. Bars: 10 μM.

**Figure 6 ijms-25-05762-f006:**
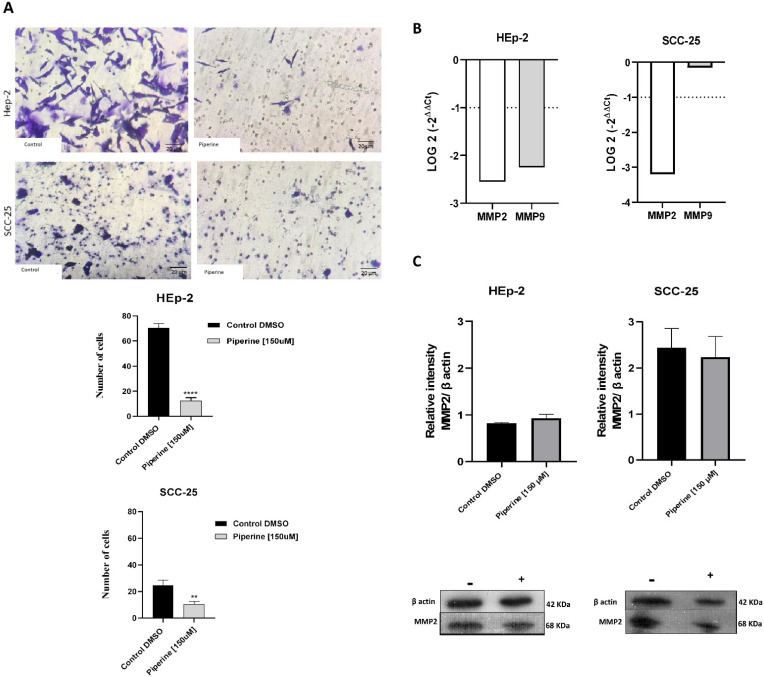
Transwell migration assay of HEp-2 and SCC-25 cells (**A**). Photomicrographs of cell migration of the control group of cells and those treated with piperine (150 μM), after 24 h, and graphs representing densitometry. Magnification of 200X, 20 μm scale. Graphs of MMP2 and MMP9 mRNA expression after piperine treatment compared to control in HEp-2 and SCC-25 cells (**B**). The dotted line (≥1.0 or ≤−1.0) is equivalent to the significant expression difference based on log 2. Graphs of MMP2 protein expression (**C**) assessed by Western blot in HEp-2 and SCC-25 cells, after treatment with piperine [150 µM], at 24 h in both assays. *n* = 3. Bars: 20 μM. ** *p* < 0.01, **** *p* < 0.0001.

**Figure 7 ijms-25-05762-f007:**
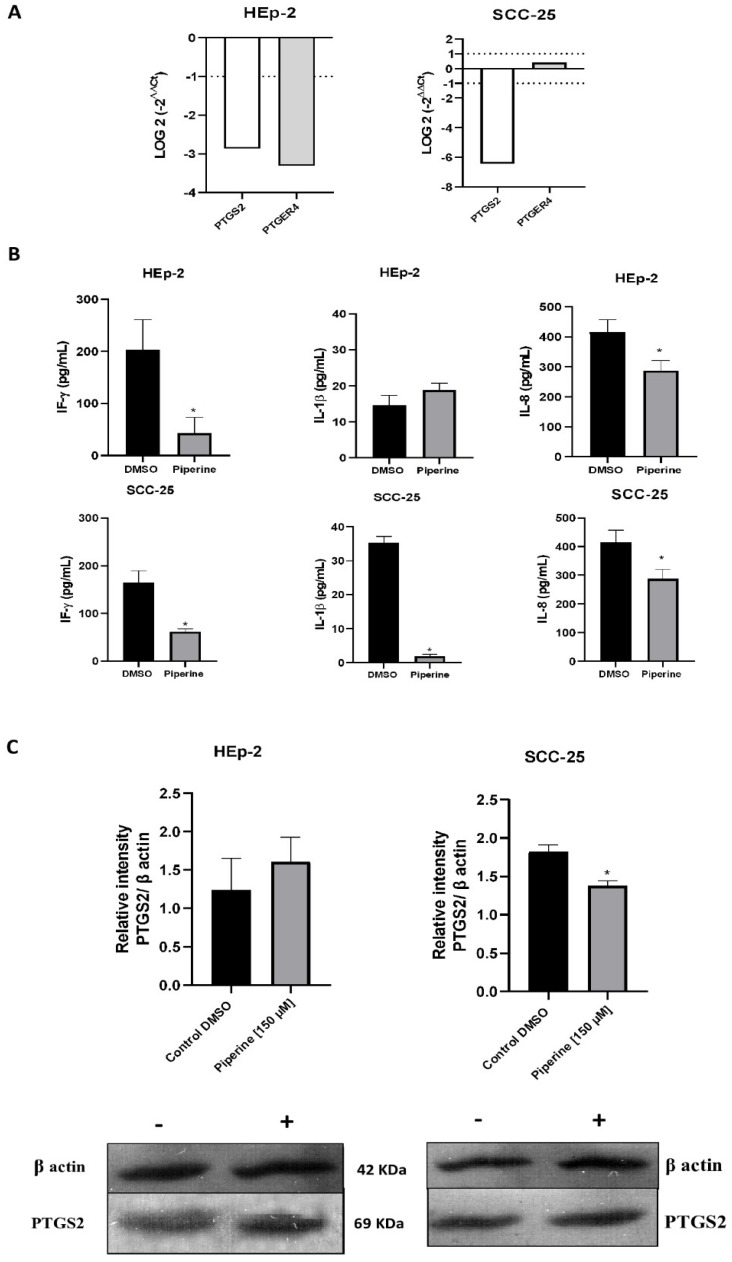
Graphs of PTGS2 and PTGER4 mRNA expression after piperine treatment compared to control in HEp-2 and SCC-25 cells (**A**). The dotted line (≥1.0 or ≤−1.0) is equivalent to the significant expression difference based on log 2. Graphs of the colorimetric ELISA assay for the analysis of cytokines/chemokines IL-8, IL-1β, IFN-γ secreted by HEp-2 and SCC-25 cells, after treatment with piperine (150 µM), at the 24-h time point in both conditions (**B**). * vs. control, *p* < 0.05. Graphs of PTGS2 protein expression (**C**) evaluated by Western blot in HEp-2 and SCC-25 cells, after treatment with piperine [150 µM], at 24 h. *n* = 3.

**Figure 8 ijms-25-05762-f008:**
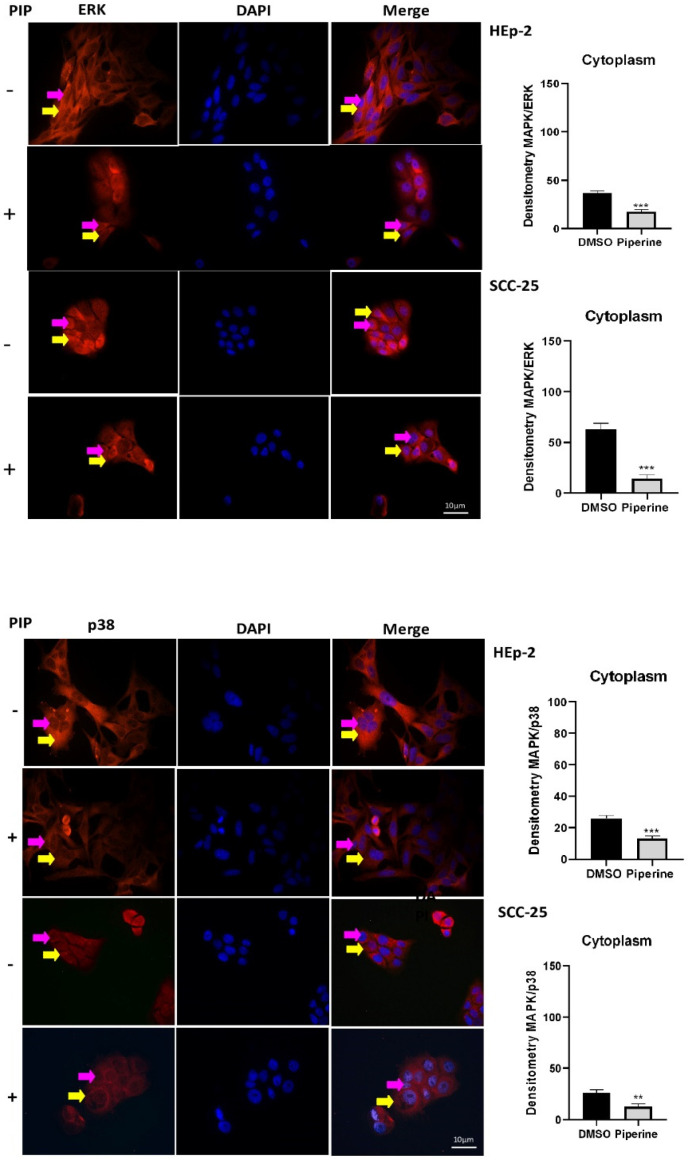
Immunocytochemistry of ERk/MAPK protein expression is indicated by a yellow arrow in the cytoplasm and a pink arrow in the nucleus. The graphs show the densitometry in each cell of the control (DMSO) and piperine [150 µM] treatment groups, at the 24-h time point. ** vs. control, *p* < 0.01; *** vs. control, *p* < 0.001. Immunocytochemistry of p38/MAPK protein expression is indicated by a yellow arrow in the cytoplasm and pink in the nucleus. The graphs show the densitometry in each cell of the control (DMSO) and piperine [150 µM] treatment groups, at the 24-h time point. ** vs. control, *p* < 0.01; *** vs. control, *p* < 0.001. *n* = 10. Bars: 10 μM.

## Data Availability

Data are contained within the article and Appendix A.

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
