# Peer review of "Molecular Aspects of Piperine in Signaling Pathways Associated with Inflammation in Head and Neck Cancer"

_ijms, 2024, doi:10.3390/ijms25115762_

Round 1
Reviewer 1 Report
Comments and Suggestions for Authors
An interesting study on the pepper alkaloid piperine with a characterization of its mild antiproliferative and proapoptotic actions in vitro, using two H&N cell lines. The activity of the natural product is very modest, not particularly interesting, but the data and experiments are correctly reported.
A few points to improve the manuscript:
1. The authors underline the DNA-binding capacity of piperine in the Discussion but this aspect is under referenced. There are recent data about this DNA binding process, to be cited (PMID: 38079271, 27995955, 26523930).
2. It is important to note that piperine is apparently a reactive molecule, susceptible to form covalent linkage with thiol-containing molecules and cysteine residues of various proteins. The product metabolism can be evoked (PMID: 31368246).
3. Recent reviews on piperine could be cited as well, to mention the multiple activities of this natural product (38405638, 38133200).
4. Piperine exhibit antiproliferative activities in vitro (this study and many others) but a demonstration of its activity in vivo is totally lacking. Therefore, the product is not very attractive and promising. The strength of the Discussion should be diminished; it is NOT a promising compound at all. However, its marked anti-angiogenic activity can be further underlined (PMID: 38036978).
5. The chemical structure of piperine (also known as bioperine) should be included. Not everyone is familiar with this natural product.
6. Abstract : investi- gate à investigate ; ex- pression à expression. There are many truncated words throughout the manuscript.
A major revision is needed.
Comments on the Quality of English LanguageOK
Author Response
Thank you very much for this review to improve our manuscript. We made a major revision and added your suggestions to make our manuscript better:
- The authors underline the DNA-binding capacity of piperine in the Discussion but this aspect is under referenced. There are recent data about this DNA binding process, to be cited (PMID: 38079271, 27995955, 26523930). –
Now we explain better the DNA binding process in the discussion, highlight in the line 456-62.
- It is important to note that piperine is apparently a reactive molecule, susceptible to form covalent linkage with thiol-containing molecules and cysteine residues of various proteins. The product metabolism can be evoked (PMID: 31368246).
We have seen from this article that Pranetha characterizes the stability and reactive metabolites of piperine and so we describe in the introduction, highlighted in yellow, in the lines 50-55. We add the chemical compounds of the piperine and write about the bioactivity too in the introduction, highlighted in yellow, in the lines 56-60.
- Recent reviews on piperine could be cited as well, to mention the multiple activities of this natural product (38405638, 38133200).
Thank you very much for this suggestion. We add this references in the introduction, highlighted in yellow, in the lines 50-55 and 95.
- Piperine exhibit antiproliferative activities in vitro (this study and many others) but a demonstration of its activity in vivo is totally lacking. Therefore, the product is not very attractive and promising. The strength of the Discussion should be diminished; it is NOT a promising compound at all. However, its marked anti-angiogenic activity can be further underlined (PMID: 38036978).
I'm sorry to disagree, but I still believe that piperine is a promising compound, but I totally agree that new studies are needed, including in vivo trials, which were not part of this study, unfortunately. Therefore, we diminish the strength of being a promising compound in the discussion and we explained the importance of new studies with animal models in the discussion, highlight in the line 496-501.
- The chemical structure of piperine (also known as bioperine) should be included. Not everyone is familiar with this natural product.
Thank you for this comment, it really is very important to insert the chemical structure of piperine, now we inserted in the introduction, lines 50-91.
- Abstract :investi- gate à investigate ; ex- pression à expression. There are many truncated words throughout the manuscript.
I am really sorry about this. There was a technical problem in our template configurations. Now the problem has been partially solved, because the journal ask me insert our manuscript in this template, but it generates these hyphens because it doesn't fit in the space that they recommend.
Reviewer 2 Report
Comments and Suggestions for Authors
In this manuscript, the authors report on the biological and molecular consequences of piperine exposure on head and neck (H/N) tumor cell lines. It has been reported in the literature that piperine has an impact on cancer cell proliferation and viability in other cancer types. The results are therefore not new, and aim to determine whether piperine might be active in H/N cancer.
It appears that piperine significantly impedes cell growth, with an impact on the ERK pathway and inflammation in vitro. Such relationships were already reported in other cancer/normal settings.
There are also no confirmatory in vivo data, e.g. xenograft experiments.
Major comments:
1) Lack of in vivo studies. Can the author provide evidence that piperine may act on H/N cancer cells in vivo ?
2) Piperine has a major impact on cell growth. How does it interfere with the interpretation of migration and colony formation assays? Is it possible to confirm the data by short-term exposure to the compound? Or by seeding a higher concentration of piperine-treated cells to see whether it compensates for the delayed growth?
3) The data show a defect in the cell cycle with a massive increase in G2 phase cells. Can the authors provide an Edu vs. DNA graph to ensure that there is no re-replication and/or mitotic slippage?
4) Molecular defects reported (MMP, PTGS and MAPK) are rather modest, or even contradictory between mRNA and protein analyses. Why are these markers examined? This needs to be better explained.
In addition, it would be important to confirm whether these changes are functional, using pharmacological inhibitors for example.
Minor :
The use of the hyphen is problematic.
Comments on the Quality of English Language
The use of the hyphen is problematic.
Author Response
Thank you very much for this review to improve our manuscript. Really the piperine impedes cell growth, with an impact on the ERK pathway and inflammation in vitro, that is innovative in head and neck cancer cells. Really there is no in vivo data, which were not part of this study, unfortunately, but it is necessary in the next step. We made a revision and added your suggestions to make our manuscript better:
1) Lack of in vivo studies. Can the author provide evidence that piperine may act on H/N cancer cells in vivo ? existem estudo in vivo que relatam essa relação da piperine mas não foi o foco do trabalho.
I agree with you that in vivo studies would increase the robustness of the results found, but unfortunately they were not part of this study. But studies explaining that piperine can act on H/N cancer in vivo were provided in the discussion, highlighted in lines 496-501.
2) Piperine has a major impact on cell growth. How does it interfere with the interpretation of migration and colony formation assays? Is it possible to confirm the data by short-term exposure to the compound? Or by seeding a higher concentration of piperine-treated cells to see whether it compensates for the delayed growth?
Thank you for the comments, now we highlight the relationship between cell growth (proliferation) and invasion in the discussion, highlighted in lines 468-71.
It was possible to confirm the data by short-term exposure from the literature and it was also possible to observe that at 48 and 72 hours the effect was almost the same with 24 hours of treatment in the proliferation assay. And we also correlated the chosen concentration and the time with data from the literature in lines 440-5.
3) The data show a defect in the cell cycle with a massive increase in G2 phase cells. Can the authors provide an Edu vs. DNA graph to ensure that there is no re-replication and/or mitotic slippage?
Unfortunately, the flow cytometer assay used in this article does not allow us do a graph to ensure whether re-replication or mitotic slippage is occurring. Other assays or other equipment would be necessary to provide these answers.
4) Molecular defects reported (MMP, PTGS and MAPK) are rather modest, or even contradictory between mRNA and protein analyses. Why are these markers examined? This needs to be better explained. In addition, it would be important to confirm whether these changes are functional, using pharmacological inhibitors for example.
Thank you for this comment, but contradictory analyses between mRNA and protein expression occur because different cellular moments (central dogma) occur, while one cell is transcribing another is translating. Of course, we expected to find the same result between mRNA and protein expression, but unfortunately for reasons specific to each cell, they may not match our expectations all the time.
About the markers, they were examined in order to better understand the mechanisms of cancer progression and they were widely studied by our research group previously, because tumorigenic cells have a characteristic of uncontrolled growth, and it is known that as arachidonic acid derivatives participate in inflammation and are also closely linked to tumor development, with PTGS2 being highly expressed in hyperplastic tissues. The study of the genes of this inflammatory pathway is necessary in head and neck cancer, then we added this explanation in the discussion, highlighted in lines 472-6.
We recognize that it would be important to confirm our findings with functional studies, using pharmacological inhibitors for example, but for financial reasons this has not been done, but in a new PhD project we will be able to do it. We added this in the discussion, highlighted in lines 505-7.
Minor :
The use of the hyphen is problematic.
I am really sorry about this. There was a technical problem in our template configurations. Now the problem has been partially solved, because the journal ask me insert our manuscript in this template, but it generates these hyphens because it doesn't fit in the space that they recommend.
Reviewer 3 Report
Comments and Suggestions for Authors
In this study, the authors investigated the effects and mechanisms of piperine on the signaling pathways in head and neck cancer. The authors concluded that piperine exhibited strong anticancer activity due to its anti-inflammatory and antiproliferative properties, and could be a prospective and integrative therapeutic option for patients with head and neck cancer.
Comments
The reviewer has some concerns as follows:
1. In the end of Introduction section, the authors described that the anticancer effect of piperine on head and neck carcinoma has yet to be elucidated. However, this description is not correct. Recently, there are study and review articles for this issue, such as Han at al. Piperine Induces Apoptosis and Autophagy in HSC-3 Human Oral Cancer Cells by Regulating PI3K Signaling Pathway. Int J Mol Sci. 2023 Sep 11;24(18):13949; Benayad et al., The Promise of Piperine in Cancer Chemoprevention. Cancers (Basel). 2023 Nov 20;15(22):5488. The effects and possible molecular mechanisms of piperine on cancers including head and neck carcinoma have been shown. Therefore, the novelty of this study is not high.
2. The data presentation of this study needs to be highly improved:
(1) In all figures, the images are not clear (the image/figure quality is poor) and the font size is too small to clear. Image resolution needs to be improved.
(2) In Figures 1E, F and 2A, the units for piperine can be shown (μM).
(3) In Figure 4, which images are control group and which are piperine-treated group?
(4) In Figures 5B and 6A, why there are no standard error (or deviation) and statistical analysis for these mRNA data?
(5) In Figure 6C, from the presented blot images, the change/difference for PTGS2 protein expression between two groups are very weak.
(6) In Figure 7, it is hard to distinguish the change/difference for ERK protein expression in cytoplasm between two groups. Moreover, why are the cell densities so low in both groups?
3. The writing of the manuscript is not very smooth and some of grammars are incorrect such as lines 319-320. It is recommended to revise it.
4. Overall, the presented results cannot support the conclusions.
Comments on the Quality of English LanguageThe writing of the manuscript is not very smooth and some of grammars are incorrect such as lines 319-320. It is recommended to revise it.
Author Response
Thank you very much for this review to improve our manuscript. We made a major revision and added your suggestions to make our manuscript better:
- In the end of Introduction section, the authors described that the anticancer effect of piperine on head and neck carcinoma has yet to be elucidated. However, this description is not correct. Recently, there are study and review articles for this issue, such as Han at al. Piperine Induces Apoptosis and Autophagy in HSC-3 Human Oral Cancer Cells by Regulating PI3K Signaling Pathway. Int J Mol Sci. 2023 Sep 11;24(18):13949; Benayad et al., The Promise of Piperine in Cancer Chemoprevention. Cancers (Basel). 2023 Nov 20;15(22):5488. The effects and possible molecular mechanisms of piperine on cancers including head and neck carcinoma have been Therefore, the novelty of this study is not high.
I am sorry about this mistake, we take out this phrase “yet to be elucidate” and change to not completely elucidated in the introduction highlighted in lines 103-5.
I would like to point out that although the novelty of this study is not so significant, we consider that treatment with piperine has already been studied in head and neck cancer, but not in these cell lines or in these specific types of laryngeal and tongue cancer and never related of this genetic pathway, for these reasons we consider our study interesting.
- The data presentation of this study needs to be highly improved:
(1) In all figures, the images are not clear (the image/figure quality is poor) and the font size is too small to clear. Image resolution needs to be improved.
(2) In Figures 1E, F and 2A, the units for piperine can be shown (μM).
(3) In Figure 4, which images are control group and which are piperine-treated group?
(4) In Figures 5B and 6A, why there are no standard error (or deviation) and statistical analysis for these mRNA data?
(5) In Figure 6C, from the presented blot images, the change/difference for PTGS2 protein expression between two groups are very weak.
(6) In Figure 7, it is hard to distinguish the change/difference for ERK protein expression in cytoplasm between two groups. Moreover, why are the cell densities so low in both groups?
Thank you very much for this appointments, now you change some points in this paper: (1)We improved all the figures, now the image quality and the resolution become better than before and the font size became bigger. (2) We add μM in the legends. (3) In Figure 4, we explain in the legend that images are from piperine-treated group. (4) In Figures 5B and 6A, we used a math model with -Ct comparing, the 2−ΔΔCT Method (Livak and Schmittgen, 2001), then we did not used statistical analysis for these mRNA data. (5) Really the difference for PTGS2 protein expression between two groups are very weak in blot, but in the densitometry, it was statistically significant. (6) In Figure 7, it is not so hard to distinguish the difference for ERK protein expression in cytoplasm, in our opinion, because the intensity of the red color is higher in the control group in both cell lines studied and densitometry shows this. A density of 50 showed in the figure 7 would not be a density of the cells, but a density of the dye and this value is not considered low for this technology, as is possible to see in another paper (Cardoso et al., 2023).
- The writing of the manuscript is not very smooth and some of grammars are incorrect such as lines 319-320. It is recommended to revise it.
Thank you for reading and correcting the english language, I am sorry about this. We've rewritten that sentence, now on lines 523-24, highlighted in yellow. But sometimes the template of the journal is making it difficult our formatting.
- Overall, the presented results cannot support the conclusions.
I'm sorry this isn't clear, but the results about antiproliferative and anti-inflammatory effects were written on lines 175 and 331, respectively, support our conclusions.
Round 2
Reviewer 1 Report
Comments and Suggestions for Authors
The revised manuscript is a little improved. My previous comments have been addressed but the manuscript remains relatively weak, with too many general comments in the conclusion.
Additional issues to consider:
1. Fig 1. is too large (almost a full page). Redraw the structures in a linear form and includes piperoyl coA. The structure shown is not piperidine. Remove “Source: (AZAM; PARK; KIM; CHOI, 2022)”. Places the legend below the figure, with a brief explanation of the biosynthetic pathway.
2. The drawing organization of the manuscript should be reworked to avoid large blank spaces (pages 7 and 11).
Author Response
Thank you very much for this review to improve our manuscript. We made some revision and added your suggestions to make our manuscript better.
About the conclusions we write that the piperine has anticancer activity due to its anti-inflammatory and antiproliferative properties, because they were the main results. And we wrote that piperine could be a prospective and integrative therapeutic option for patients with head and neck cancer, because these are the future perspectives, not mine only, but based in the literature too.
Additional issues to consider:
- Fig 1. is too large (almost a full page). Redraw the structures in a linear form and includes piperoyl coA. The structure shown is not piperidine. Remove “Source: (AZAM; PARK; KIM; CHOI, 2022)”. Places the legend below the figure, with a brief explanation of the biosynthetic pathway.
I am sorry about this. Now we change the figure 1 and the legend too, in the lines 57-63 and add a new reference, highlighted in yellow.
- The drawing organization of the manuscript should be reworked to avoid large blank spaces (pages 7 and 11).
Thank you for this comment, the large blank spaces were deleted.
Reviewer 2 Report
Comments and Suggestions for Authors
I thank the authors for their responses to my comments. i note that several points are not addressed by new experiments. In vivo or more detailed mechanistic/descriptive data would reinforce the overall message of the study and clarify the use of piperine in the clinic. Nonetheless, the manuscript reports important information that can be published.
Comments on the Quality of English LanguageNew references are not correctly formatted.
Author Response
Thank you very much for this review to improve our manuscript. We made some revision and added your suggestions to make our manuscript better.
- I am sorry about the new reference was not correctly formatted. We have now corrected the new reference, highlighted in yellow in the lines 693-4.
- We agree that new experiments in vivo would reinforce the use of piperine in clinical studies, then we insert one reference about this in the discussion, highlighted in lines 395-7.
Reviewer 3 Report
Comments and Suggestions for Authors
In this study, the authors investigated the effects and mechanisms of piperine on the signaling pathways in head and neck cancer. The authors concluded that piperine exhibited strong anticancer activity due to its anti-inflammatory and antiproliferative properties, and could be a prospective and integrative therapeutic option for patients with head and neck cancer.
Comments
This revised manuscript has a great improvement. The reviewer has only some minor concerns as follows:
1. In Figures 2-8, please added the information for n number in the legends.
2. In Figure 6B, please unify the colors in the bars of both Hep-2 and SCC-25 cells for MMP2 and MMP9 mRNA.
Author Response
Thank you very much for this review to improve our manuscript. We made some revision and added your suggestions to make our manuscript better.
- In Figures 2-8, please added the information for n number in the legends.
Now we add the number of experimental replicates in the legends of the Figure 2-8, highlighted in yellow.
- In Figure 6B, please unify the colors in the bars of both Hep-2 and SCC-25 cells for MMP2 and MMP9 mRNA.
I am sorry about this, now we unify the colors of the bars in the Figure 6B.